# Circulating Tumor DNA: The Dawn of a New Era in the Optimization of Chemotherapeutic Strategies for Metastatic Colo-Rectal Cancer Focusing on *RAS* Mutation

**DOI:** 10.3390/cancers15051473

**Published:** 2023-02-25

**Authors:** Shohei Udagawa, Akira Ooki, Eiji Shinozaki, Koshiro Fukuda, Kensei Yamaguchi, Hiroki Osumi

**Affiliations:** Department of Gastroenterology, Cancer Institute Hospital, Japanese Foundation for Cancer Research, Tokyo 135-8550, Japan

**Keywords:** circulating tumor DNA, liquid biopsy, colorectal cancer, *RAS* mutation

## Abstract

**Simple Summary:**

Molecularly targeted therapies have greatly contributed to the development of colorectal cancer treatments. Genomic profiling to identify gene alterations is a rapidly developing field. Liquid biopsies have recently drawn considerable attention because they offer several advantages over tissue biopsies and can be used to detect several soluble factors, including circulating tumor DNA. In this review, we discuss the usefulness of analyzing circulating tumor DNA to design more personalized and effective cancer treatments and discuss several ongoing clinical trials that aim to evaluate its utility. Genomic profiling using circulating tumor DNA could be integrated into clinical strategies for cancer treatment in the near future.

**Abstract:**

Genotyping of tumor tissues to assess *RAS* and *BRAF* V600E mutations enables us to select optimal molecularly targeted therapies when considering treatment strategies for patients with metastatic colorectal cancer. Tissue-based genetic testing is limited by the difficulty of performing repeated tests, due to the invasive nature of tissue biopsy, and by tumor heterogeneity, which can limit the usefulness of the information it yields. Liquid biopsy, represented by circulating tumor DNA (ctDNA), has attracted attention as a novel method for detecting genetic alterations. Liquid biopsies are more convenient and much less invasive than tissue biopsies and are useful for obtaining comprehensive genomic information on primary and metastatic tumors. Assessing ctDNA can help track genomic evolution and the status of alterations in genes such as *RAS*, which are sometimes altered following chemotherapy. In this review, we discuss the potential clinical applications of ctDNA, summarize clinical trials focusing on *RAS*, and present the future prospects of ctDNA analysis that could change daily clinical practice.

## 1. Introduction

Therapeutic drugs for metastatic cancers are chosen based on the organ of origin. Cancer treatment has advanced remarkably over the last few decades, and the survival rate has improved significantly. Molecular targeted therapies have contributed greatly to the development of cancer treatments. Advances in genomic profiling have enabled the identification of genetic alterations that cause cancer and have supported the development of personalized cancer treatments for each gene alteration. Frequently mutated genes in non-hypermutated colorectal cancer (CRC) include *APC*, *TP53*, *KRAS*, *PIK3CA*, *FBXW7*, *SMAD4*, *TCF7L2*, and *NRAS* [1]. The *RAS* family is one of the most frequently mutated gene families and has been extensively studied in metastatic CRC (mCRC). *RAS* is one of the major proteins involved in the mitogen-activated protein kinase (MAPK) signaling cascade. The *RAS* oncogene family includes *KRAS*, *NRAS*, and *HRAS*. The most common *RAS* mutations occur in *KRAS*; approximately 40% of CRC cases have *KRAS* mutations, while *HRAS* mutations are rarely detected [1]. *KRAS* and *NRAS* mutations are negative predictive factors for the efficacy of anti-epidermal growth factor receptor (anti-EGFR) monoclonal antibodies (mAbs), which act as both primary and secondary resistance markers [2,3]. *RAS* mutations are associated with poor prognosis in advanced stages [4]. Although it seems likely that constitutive activation of the *RAS* signaling pathway is involved in tumor progression, the reason for poor prognosis is still not well understood [5]. *BRAF* also acts as a downstream *RAS* effector in the MAPK signaling cascade. The *BRAF* V600E mutation, which is found in 8–10% of patients with mCRC, is also a negative predictive factor for the efficacy of anti-EGFR mAbs [6,7,8]. Therefore, analysis of genetic alterations is increasingly important for developing personalized cancer treatments. Tissue biopsy is extensively used to determine suitable therapeutic drugs based on the molecular profile of an individual. However, tissue biopsy has several limitations, including potential complications, difficulty in performing repeated biopsies due to the invasiveness of the procedure, difficulties in collecting tissue, and intra- and inter-tumor heterogeneity. Liquid biopsy can overcome these limitations because it is less invasive and involves the collection of bodily fluids such as blood and urine. Therefore, it allows for the repeated analysis of gene alterations over time. Liquid biopsies, including those performed for circulating tumor DNA (ctDNA), are clinically used to detect *RAS* and *BRAF* mutations and perform comprehensive genomic profiling in CRC [9,10]. This review summarizes liquid biopsy and ctDNA analyses and their applications to CRC in clinical trials.

## 2. Liquid Biopsy

Tissue examination is necessary for cancer diagnosis and management. Histological analysis can reveal the genetic profile of the tumor and enable a more accurate prognosis and prediction of systemic chemotherapy efficacy. However, while tissue biopsy is necessary for the development of a therapeutic strategy, it is invasive with potential complications such as bleeding, pain, infections, and neuropathy, and it can be difficult to obtain tissue samples due to tumor volume or anatomical reasons [11]. The extraction of tumor tissues is sometimes required after the onset of resistance, and tissue biopsies may be difficult to repeat for several reasons. In addition to safety, the number of tumor cells obtained can vary. Fine needle aspiration or core needle biopsies often result in the extraction of less tumor tissue for molecular analysis [12]. Moreover, tissue biopsies are affected by tumor heterogeneity. Metastatic tumors could have different genetic profiles, even if they were derived from a primary tumor within the same patient [13]. When the treatment decision is based on a single biopsy, intra-tumor heterogeneity can lead to therapeutic failure [14].

As an alternative to traditional tissue biopsy, liquid biopsy is useful for cancer diagnosis. Liquid specimens are obtained using minimally invasive techniques and can be used to detect several soluble factors related to tumor genetics, such as cell-free DNA (cfDNA), ctDNA, circulating tumor cells, and exosomes. Cancer-associated genetic alterations, such as point mutations, copy number variations, amplification, rearrangements, aneuploidy, and fusion and methylation patterns, have been detected in ctDNA [12]. Cancer patients have higher levels of plasma cfDNA than tumor-free patients; however, high levels of cfDNA are not specific to cancer [15]. Confounding factors that can contribute to the release of cfDNA include smoking, pregnancy, exercise, and numerous non-malignant disorders, such as inflammatory conditions, anemia, heart disease, metabolic syndrome, and autoimmune disorders [16].

Compared to tissue biopsy, liquid biopsy is minimally invasive and allows repeated analyses over the course of treatment for the dynamic monitoring of molecular changes in the tumor. Furthermore, liquid biopsy can overcome difficulties related to both intra- and inter-tumor heterogeneity.

## 3. ctDNA

First reported in 1948 by Mandel and Metais [17], cfDNA is used in prenatal assessments. The size distribution of cfDNA in the plasma of pregnant women ranges from 160 bp to 180 bp [18]. Increased cfDNA levels in the blood of patients with various types of cancer were first reported in 1977 [19]. Moreover, cfDNA may be released from healthy, inflamed, or diseased tissues where cells are undergoing apoptosis or necrosis and is detected in body fluids such as blood, urine, cerebrospinal fluid, pleural fluid, ascites, and saliva [20,21,22,23,24]. In patients without cancer, cfDNA concentrations range from 0–100 ng/mL (mean, 13 ± 3 ng/mL). In contrast, the mean cfDNA concentration in patients with cancer is 180 ± 38 ng/mL [19]. ctDNA is a DNA fragment derived specifically from a tumor, thus differentiating it from cfDNA. The difference in DNA concentration between patients with and without cancer is reflective of the ctDNA derived from cancer cells.

The half-life of ctDNA is approximately 2 h [25], while that of protein biomarkers is several days [26]. Therefore, with ctDNA, real-time changes in the genetic status of the tumor can be evaluated, which is in contrast to commonly used protein biomarkers such as carcinoembryonic antigens. ctDNA can quickly reflect changes in tumor burden following surgery and chemotherapy and can be used to predict disease progression and recurrence. Protein markers are not involved in tumorigenesis, whereas genetic alterations detected in ctDNA are generally the cause of tumorigenesis. Therefore, ctDNA is a more sensitive and specific biomarker than protein biomarkers because it reflects genetic alterations derived from tumors in real-time [27]. Compared to tissue biopsy, ctDNA may have the advantage of a short turn-around time (TAT), which is defined as the number of days between the test order date and the report date [28]. ctDNA genotyping significantly shortened the screening duration in SCRUM-Japan GOZILA, an observational ctDNA-based screening study that evaluated the utility of ctDNA in patients with advanced gastrointestinal cancer [29]. In CRC, the median TAT when detecting ctDNA using comprehensive ctDNA testing with the Guardant360^®^ assay was significantly lower than that for tissue testing when the complete process from sample acquisition to results was considered [30].

High analytical sensitivity and specialized equipment are required for the detection of ctDNA. Current techniques used for the quantification of tumor-associated genetic alterations can result in false-negative results; however, concordance between tissue and plasma tests for ctDNA is generally high in CRC [9,31]. Both tissue and plasma tests for ctDNA sometimes yield false-negative and false-positive results, but they can be more reliable when used in combination [32,33]. Utilizing ctDNA analysis alongside tissue tests increases the identification of biomarkers by 19.5% because it allows identification even without conclusive tissue results due to tissue insufficiency, test failure, or false negatives [30].

The amount of ctDNA in an individual is lower than that of cfDNA, and it is sometimes as low as 0.01–1.70% in curable CRC [34]. This makes it difficult to detect and quantify ctDNA with the sensitivity required for meaningful clinical use. The amount of ctDNA produced is influenced by several factors. ctDNA detection depends on the ctDNA shedding rate per cancer cell, but this can vary by multiple magnitudes between patients. Thus, the discordance between plasma- and tissue-based analyses could be due to low ctDNA shedding from tumors; the median variant allele frequency (VAF) between concordant and discordant cases was statistically different [9]. In addition, ctDNA levels reflect the total systemic tumor burden and size [25,35]. ctDNA levels decrease after complete surgery or in response to chemotherapy and generally increase with disease progression before radiological examination. Furthermore, the ctDNA detection rate varies for each organ. Because of the difficulty in detecting ctDNA, the detection rate of *RAS* mutations is low in patients with CRC with lung and peritoneal metastases [9,36]. This may be caused by differences in the distribution of DNAase depending on the metastatic site [37]. ctDNA is detected in >75% of patients with advanced pancreatic, ovarian, colorectal, bladder, gastroesophageal, breast, melanoma, hepatocellular, and head and neck cancers, but it is detected in less than 50% of patients with primary brain, renal, prostate, or thyroid cancers. In addition, ctDNA from neoplasms confined to the central nervous system and mucinous neoplasms is frequently undetectable [38]. Circulating tumor cells can also release ctDNA and therefore influence the detection of ctDNA [39].

Measuring ctDNA levels could be confounded by biological signaling arising from somatic mosaicism. Clonal hematopoiesis (CH) is a somatic mosaicism resulting from the accumulation of somatic mutations in hematopoietic stem cells. CH is influenced by age, prior radiation therapy, chemotherapy, and tobacco use and can be detected by ctDNA analysis. However, we must consider the possibility that CH can be interpreted, incorrectly, as a mutation [40,41]. In CRC, *TP53*, *GNAS*, *PTEN*, and *KRAS* mutations have been reported as CH; however, the complete distinction between tumor-derived mutations and CH is difficult to achieve [42,43].

### 3.1. Approaches for ctDNA Detection

Several techniques exist for evaluating ctDNA; however, these techniques require high sensitivity because of the low amount of ctDNA. Liquid biopsy analyses are available and include polymerase chain reaction (PCR)- and next-generation sequencing (NGS)-based platforms. 

PCR-based assays can only detect targets with known driver mutations, and they fail to detect complex genomic alterations. Highly sensitive PCR-based assays, such as droplet digital PCR (ddPCR) and beads, emulsion, amplification, and magnetics (BEAMing), have been developed [44]. ddPCR is a highly sensitive and accurate quantification method that detects low-frequency variants by amplifying single DNA molecules. Amplicon sequencing and hybridization capture reduces the background error rates of sequencing [45]. The limit of detection for ddPCR is 0.01–0.10% [46]. BEAMing is a PCR-based technique that uses flow cytometry to detect ctDNA [47]. The OncoBEAM^TM^ RAS CRC Kit, which detects 34 mutations in *KRAS*/*NRAS* codons 12, 13, 59, 61, 117, and 146, is a platform for detecting *RAS* mutations in the plasma using BEAMing technology. The OncoBEAM^TM^ RAS CRC Kit detects alterations at a 0.01% allele frequency [47]; it received market approval on 1 July 2019, from Japan’s Ministry of Health, Labour, and Welfare, and it has been covered under insurance since 1 August 2020. In a study comparing four commercial platforms that detect *KRAS*/*NRAS* ctDNA mutations, BEAMing exhibited higher sensitivity than the Idylla^TM^ KRAS Mutation Test, ddPCR, and NGS [48,49,50]. There is a high degree of concordance, of 86.4–93.3%, between ctDNA analysis with BEAMing and tissue analysis [9,51,52,53]. However, discordance was observed between plasma and tissue analyses employing BEAMing of *RAS* mutations associated with lung metastasis [9]. Other factors associated with discordance include peritoneal metastasis, mucinous carcinoma type, administration of treatment prior to liquid biopsy, longest diameter, and lesion number. Due to high concordance, we do not have to consider the cutoff for patients with only liver metastases; however, we need to consider the cutoff when patients have peritoneal metastases alone with a lesion diameter <20 mm, lung metastases alone with a lesion diameter <20 mm, or <10 lesions in total [9,36,53]. Therefore, caution should be exercised when assessing *RAS* mutations with BEAMing.

NGS can be used to analyze a large number of genes (hundreds to thousands). NGS is designed to detect multiple classes of genetic alterations, including indels, rearrangements, and copy number alterations, in both known and unknown driver genes [54]. NGS is limited by its relatively low sensitivity and high cost; however, the last decade has witnessed improvements in NGS in terms of reliability and cost [55,56]. The Guardant360^®^ assay (Guardant Health, Inc., Redwood City, CA, USA) and FoundationOne^®^ Liquid (Foundation Medicine, Cambridge, MA, USA.) are among the most popular NGS-based ctDNA testing methods. FoundationOne^®^ Liquid received market approval on 22 March 2021, from Japan’s Ministry of Health, Labour, and Welfare and has been covered under insurance since 1 August 2021. Caris Assure^TM^ liquid biopsies, whole exome DNA sequencing, and whole transcriptome RNA sequencing are comprehensive tumor profiling technologies that include all 22,000 genes. This comprehensive approach identifies cancer biomarkers and assesses the molecular features of the patient using circulating nucleic acid sequencing, which is a novel molecular profiling approach that analyzes cfDNA, cell-free RNA, genomic DNA, and RNA from circulating white blood cells [57].

Methods for evaluating ctDNA are divided into tumor-informed and tumor-agnostic, with the previously mentioned techniques for evaluating ctDNA being tumor-agnostic (Appendix A). The tumor-informed approach requires genomic profiling of the tumor tissue. This approach identifies the alterations derived from tumors. However, the tumor-agnostic approach does not require the mutational status of tumor tissue and is based on panel-based sequencing. Many studies based on a tumor-informed approach have been conducted. The advantages of the tumor-informed approach include personalized analysis and accuracy in tracking the molecular responses [58]. Signatera^TM^ is a novel, highly sensitive, and specific approach for ctDNA detection. This is a personalized and tumor-informed approach for minimal residual disease (MRD) assessment. A primary tumor sample was used for whole-exome sequencing to assess the differences in over 20,000 genes between a patient’s tumor sample and a normal DNA sample. Sixteen highly ranked patient-specific mutations were selected for the panel. The samples are amplified using a patient-specific assay, barcoded, pooled, and sequenced using an NGS platform. Somatic alterations derived from tumors are then detected in the plasma [59]. Signatera^TM^ has been useful in assessing MRD in recent clinical studies. Personalized approaches have advanced with the development of methods to detect cancer-specific genomic alterations. 

### 3.2. Assessment of Prognosis

Cancer prognosis is assessed based on clinical observations, tumor type, staging, and histopathological and biomolecular characterization. The amount of ctDNA could be a prognostic factor (Appendix A) [60,61,62,63,64,65,66]. In the CORRECT trial, a retrospective exploratory analysis evaluating the efficacy and safety of regorafenib in mCRC patients, high baseline *KRAS* mutant allele frequency (MAF) and circulating DNA concentrations were associated with a shorter median overall survival (OS) [60]. ctDNA measured using VAF at baseline was a prognostic factor potentially related to initial tumor volume in patients with *RAS* wild-type (WT) mCRC who were eligible for initial therapy with panitumumab plus FOLFOX (fluorouracil, leucovorin, and oxaliplatin) [61]. In addition, several reports have revealed an association between the amount of baseline ctDNA, VAF or MAF and prognosis in mCRC patients [62,63,64,65,66]. ctDNA methylation markers are gaining attention for the diagnosis and prognosis of CRC. A model using five selected cfDNA methylation markers was useful as an independent prognostic risk factor in multivariate analysis [67]. In addition, *PIK3CA* mutations at baseline are associated with poor outcomes in patients with *RAS* WT mCRC [68]. ctDNA analysis may allow us to obtain information on the factors influencing prognosis.

### 3.3. Detection of Recurrence: Minimal Residual Disease

The early detection of micrometastatic lesions that are undetectable by radiological examination is essential to reduce the risk of incurable metastasis. Recurrence was monitored following cancer treatment. ctDNA is sufficiently sensitive for the detection of MRD following surgical resection [35]. Positive ctDNA detection in resected early-stage colon cancer (CC) precedes the radiological detection of recurrence by more than a few months [69,70]. ctDNA analysis following surgery is a promising prognostic assessment and can aid in the identification of patients with a very high risk of recurrence [71,72,73], which could reduce unnecessary chemotherapy. A ctDNA-guided approach for treating pathological stage II CC reduces adjuvant chemotherapy (AC) without compromising recurrence-free survival [74]. Patients who did not previously require AC may benefit from AC if ctDNA predicts recurrence.

A prospective, multicenter cohort study indicated an association between ctDNA and recurrence in patients with stage I–III CRC following curative surgery. Following curative surgery, 10.6% of patients tested positive for ctDNA on postoperative day 30. Notably, ctDNA-positive patients were seven times more likely to relapse than ctDNA-negative patients (hazard ratio (HR), 7.2; 95% confidence interval (CI), 2.7–19.0; *p* < 0.001). Moreover, seven patients who were positive for ctDNA following AC had recurrences. Among 75 patients with longitudinally collected plasma samples, 14 of 15 ctDNA-positive patients experienced recurrence compared to 2 of 60 ctDNA-negative patients. ctDNA indicative of recurrence was detected 8.7 months earlier than the diagnosis using standard-of-care radiologic imaging [69]. Similar results have been reported in breast, lung, and bladder cancer [75,76,77]. Thus, ctDNA could help clinicians decide to pursue more intense therapy in patients with a higher risk of recurrence. ctDNA can be used to stratify patients according to their risk of recurrence, enabling therapeutic intervention before the development of clinical metastasis.

In CIRCURATE-Japan, three clinical trials using Signatera^TM^ are ongoing to evaluate the clinical benefits of ctDNA and refine AC for CRC. The GALAXY study was designed to monitor ctDNA status in stage II–IV patients who were eligible for curative surgery [78]. ctDNA was analyzed before surgery and at 4, 12, 24, 36, 48, 72, and 96 weeks after surgery. The VEGA trial (jRCT1031200006) is a randomized phase III study to evaluate CAPOX therapy as an AC for high-risk stage II or low-risk stage III CC patients with ctDNA-negative status four weeks after curative surgery. Patients were randomized in a 1:1 ratio to receive either CAPOX therapy for three months or surgery alone. AC may not be required after curative surgery in patients with ctDNA-negative CC. The ALTAIR trial (NCT04457297) is a randomized, phase III study to evaluate preemptive trifluridine/tipiracil therapy in patients with CC who are positive for ctDNA after curative surgery for up to 2 years. The BESPOKE study (NCT04264702) examined the effect of Signatera^TM^ use on AC decisions.

The benefit of preemptive therapy in patients who are positive for ctDNA before radiologic imaging is unknown. Several clinical trials on AC using ctDNA analysis are currently ongoing. The DYNAMIC-II (ACTRN12615000381583), MEDOCC-CrEATE (NL6281/NTR6455), COBRA (NCT04068103), and IMPROVE-IT (NCT03748680) trials are investigating the administration of AC depending on ctDNA levels in patients with stage I or II CRC. In these trials, ctDNA-positive patients receive AC or follow-up if ctDNA is negative. The PEGASUS (NCT04259944) and DYNAMIC-III (ACTRN12617001566325) trials including resected stage III or T4N0 stage II CC are also ongoing. Both trials are investigating ctDNA-guided AC. ctDNA-negative patients will receive de-escalated AC and ctDNA-positive patients will receive escalated AC. OPIMIZE (NCT04680260) is an open-label, randomized phase II trial for patients receiving radical treatment for metastatic spread of CRC. Patients were randomized between the standard-of-care and ctDNA-guided treatments. ctDNA-positive patients receive FOLFOXIRI (fluorouracil, leucovorin, oxaliplatin, and irinotecan), and ctDNA-negative patients receive capecitabine or observation only. The results from these trials could shift AC from a conventional strategy to a ctDNA-guided strategy.

### 3.4. Predicting Response to Treatment and Monitoring Acquired Resistance 

Genetic alterations in tumor DNA are important markers for deciding the treatment regimen and predicting the response to treatment. High levels of *KRAS* mutant (MT) alleles in the plasma are a clear indicator of response to treatment in metastatic CC [79,80]. ctDNA allows the design of specific treatments according to genetic alterations. Combinations of *BRAF* inhibitors and anti-EGFR mAbs are effective in mCRC patients harboring a *BRAF* V600E mutation. The use of ctDNA analysis to detect *BRAF* V600E offers an opportunity to administer *BRAF* inhibitors. ctDNA can be used to detect *BRAF* V600E in the plasma of patients in whom it was not detected in tissue analysis due to spatial heterogeneity [81]. Microsatellite instability (MSI) is associated with a higher risk of cancer and has been assessed in solid tumors. MSI-high tumors are sensitive to immune checkpoint inhibitors (ICIs). MSI is typically assessed in tumor tissues using immunohistochemistry and PCR-based assays. Recently, a high concordance rate between MSI measured using conventional tissue and ctDNA-based approaches has been reported. In the near future, ctDNA-based approaches to detect MSI could be used to identify patients who could benefit from ICIs [82]. In melanoma, assessment of the ctDNA baseline could indicate clinical outcomes in patients receiving ICI treatment [83,84,85]. In patients with mCRC with *HER2* amplification, a combination of pertuzumab and trastuzumab may be effective. The TRIUMPH trial is a prospective phase II study involving mCRC patients with *HER2* amplification and investigating pertuzumab and trastuzumab as a second-line or later treatment. Twenty-five patients with *HER2* amplification, confirmed using ctDNA, received pertuzumab plus trastuzumab and achieved an overall response rate (ORR) of 28%, compared to 30% in 27 tissue-positive patients [86]. In the HERACLES trial, which evaluated trastuzumab plus lapatinib or pertuzumab plus trastuzumab-emtansine in patients with *HER2*-positive mCRC, comprehensive ctDNA analysis identified that more than 85% of patients showed primary resistance when treated with lapatinib and trastuzumab [87]. Moreover, an adjusted *ERBB2* plasma copy number has been correlated with progression-free survival (PFS) and best objective response [88]. Therefore, ctDNA analysis could be useful for identifying patients who would benefit from *HER2* blockade. ctDNA analysis could provide relevant molecular profiling required for tumor-agnostic treatment. *NTRK* fusion is an oncogenic driver that is also present in CRC. The FDA granted tumor-agnostic approval to the TRK inhibitors larotrectinib and entrectinib. Tissue testing is routinely used to detect *NTRK* fusions; however, these fusions can also be detected using ctDNA with a high positive predictive value [89].

Monitoring tumor responses using ctDNA during the course of treatment could allow for changes in the drugs administered before the observation of disease progression on radiological examination. A small or absent early decrease in ctDNA levels during mCRC treatment was associated with short PFS and OS in a systematic review and meta-analysis [90]. Targeted therapies are effective for specific genetic mutations; however, most patients eventually develop secondary resistance. Designing the next treatment requires the identification of the mechanism of acquired resistance. The advantages of liquid biopsies include the simplicity of specimen collection and the ability to provide snapshots to detect the emergence of resistant clones. Almost all patients with CRC acquire resistance to *KRAS* mutation inhibition during anti-EGFR mAb therapy [91,92]. Multiple alterations conferring resistance to anti-EGFR mAbs, other than *RAS* mutations, were also observed. The cfDNA profiles of 42 patients with EGFR extracellular domain (ECDs) mutations, which are implicated in acquiring resistance to anti-EGFR mAbs, harbor *MEK1* and *BRAF* mutations and *KRAS*, *MET*, *ERBB2*, and *KIT* amplifications [93,94]. *FLT3* amplification and *MAP2K1* are resistant to anti-EGFR mAbs [95]. Multiple alterations were observed in most cases. Predictive markers for sensitivity to anti-EGFR mAbs include *RAS* and several other alterations. In patients with *HER2* blockade in CRC, the emergence of resistance alleles such as *PIK3CA* is observed, which indicates that they might be sub-clonal [88]. In addition, some patients showed clear progression in one lesion, whereas the response was stable in the other. This indicates heterogeneity within a single patient [88]. Acquired resistance has also been observed against *BRAF*-targeted therapy in the form of *NRAS* and *MEK1/2* mutations, *BRAF* amplification, or *CRAF* overexpression [96]. ctDNA can be used to track the emergence of resistant clones throughout the course of treatment because of the accessibility of plasma from patients. Understanding the mechanisms underlying acquired resistance to treatment could lead to improved personalized anticancer therapy and the development of combinatorial treatment strategies. Changes in resistance to anticancer therapy are not fully understood. Comprehensive gene profiling rather than single molecular evolution should be used to understand drug resistance.

## 4. Anti-EGFR Monoclonal Antibody Rechallenge

The concept of anti-EGFR mAb rechallenge was first reported by Santini et al. [97]. They assessed the activity of cetuximab rechallenge in patients with mCRC. The results were promising, with a response rate (RR) of 53.8% and median PFS (mPFS) of 6.6 months. However, they did not distinguish between anti-EGFR mAbs rechallenge and reintroduction. Rechallenge is defined as anti-EGFR mAb re-administration after an anti-EGFR-mAb-free period in patients with prior resistance to anti-EGFR mAbs. Reintroduction, on the other hand, is defined as anti-EGFR mAb re-administration after prior anti-EGFR discontinuation in patients without resistance. Additionally, an intermittent anti-EGFR mAb strategy has been proposed. Recently, it was reported that intermittent panitumumab, instead of continuous panitumumab with chemotherapy, produced a long PFS with reduced skin toxicity [98]. Thus, some similar strategies have been proposed for anti-EGFR mAbs. The development of these treatment strategies requires the elucidation of resistance mechanisms.

Acquired resistance to anti-EGFR mAbs is associated with the emergence of *RAS* mutations [91,92]. *RAS* mutations are likely to be present at undetectable levels before the administration of anti-EGFR mAbs; the number of *RAS* MT cells increases to a detectable level during the administration of anti-EGFR mAbs [99]. Second-line therapy without anti-EGFR mAbs, however, causes the reduction or disappearance of *RAS* MT subclones. One hypothesis to explain this observation is that targeted therapies apply selective pressure on heterogeneous tumors, including the undetectable *RAS* MT populations, and resistant cells survive; however, these resistant cells may be innately limited [100]. Therefore, *RAS* mutations may exist as sub-clonal mutations with low allele frequencies. The half-life of these mutations is approximately 3–4 months after withdrawal of anti-EGFR mAbs [100]. This could restore sensitivity to anti-EGFR mAbs [53]. The disappearance of *RAS* mutations could be attributed to a decline in the percentage of acquired mutated *RAS* alleles to below the limit of detection during treatment without anti-EGFR mAbs [95]. Tracking the dynamics of resistant sub-clonal populations allows the identification of patients who can be rechallenged with the same drugs. There is a rationale for anti-EGFR mAb rechallenge following failure of second-line treatment with an anti-EGFR-mAb-free therapeutic window (Figure 1). We analyzed current knowledge regarding anti-EGFR mAb rechallenge based on ctDNA.

### 4.1. Trials of anti-EGFR Monoclonal Antibody Rechallenge

The prognosis of mCRC after progression to first and second therapies is poor. Third- or later-line therapies include trifluridine/tipiracil (+ bevacizumab) or regorafenib. Both therapies were clinically beneficial in a phase III clinical trial compared to the best supportive care. However, their efficacy is limited; the mPFS is approximately 2 months, and the ORR is approximately 1–4% [101,102,103,104]. In addition, toxicities, such as gastrointestinal toxicity and hematologic toxicity, should be assessed because of adverse events (AEs) frequently caused by trifluridine/tipiracil (+ bevacizumab). In addition, regorafenib causes hand–foot syndrome, fatigue, diarrhea, and hypertension. Therefore, anti-EGFR mAb rechallenge is expected to become a common new therapeutic strategy with a high response rate.

Trials of anti-EGFR mAb rechallenge and the analysis results based on liquid biopsies have been reported (Table 1). The CRICKET trial is a multicenter phase II trial for assessing the activity of cetuximab rechallenge as a third-line treatment for patients with *RAS* and *BRAF* WT mCRC who benefitted from first-line cetuximab- and irinotecan-based treatment, with at least a partial response (PR) and a PFS of at least 6 months, and then became resistant. The time between the end of first-line therapy and the start of third-line therapy was ≥4 months. Liquid biopsies for ctDNA analysis were performed at the baseline. The ORR was 21%. Four patients who achieved confirmed PR had no *RAS* mutations. However, eight patients with *RAS* MT ctDNA had stable disease (SD) or progressive disease. Patients with *RAS* WT ctDNA had significantly longer PFS than those with *RAS* MT ctDNA in a retrospective analysis (mPFS, 4.0 vs. 1.9 months; HR, 0.44; 95% CI, 0.18–0.98; *p* = 0.03). A similar trend was observed for OS [105]. An E-challenge trial, a multicenter phase II study, evaluated whether there is a correlation between the anti-EGFR-mAb-free interval (aEFI) and efficacy. Patients with an aEFI ≥ 16 weeks between the last dose of cetuximab (during previous treatment) and the start of the cetuximab rechallenge were included. Other criteria included *RAS* WT and complete response (CR), PR, or SD that persisted for 6 months or more for anti-EGFR mAb. The primary endpoint was the ORR. The ORR was 15.2%, and PR was observed in all patients. There was no statistically significant difference in ORR, PFS, or OS stratified using the median aEFI (311 days). However, in the additional liquid biopsy for ctDNA, the RR for *KRAS* G12/G13/A59/Q61, *BRAF* V600E, and *EGFR* S492R mutants increased; the RR of patients with all WT was 25% compared to 12.5% in those with any of the mutations [106]. A post hoc biomarker study (JACCRO CC-08/09AR) was performed to evaluate the association between survival outcomes and *RAS* status in ctDNAs. The JACCRO CC-08 and 09 trials evaluated the efficacy and safety of anti-EGFR mAb rechallenge as a third-line therapy for patients with *KRAS* WT mCRC who achieved a clinical response to first-line therapy with anti-EGFR mAbs. The *RAS* status was evaluated using the OncoBEAM *RAS* CRC Kit. *RAS* status in ctDNA is associated with clinical outcomes in patients with mCRC receiving anti-EGFR mAb rechallenge. Patients with *RAS* mutation at baseline had significantly shorter PFS and OS than those without *RAS* mutation (mPFS, 2.3 vs. 4.7 months; HR, 6.2; *p* = 0.013 and mOS, 3.8 vs. 16.0 months; HR = 12.4; *p* = 0.0028). The disease control rate was 80% in patients with no *RAS* mutations and 33.3% in patients with *RAS* mutations [107]. CAVE is a phase II single-arm trial to assess the efficacy of cetuximab rechallenge plus avelumab in patients with *NRAS* and *KRAS* WT mCRC who achieved CR or PR to first-line therapy with anti-EGFR mAbs. Avelumab, an immune checkpoint inhibitor, exhibits antibody-dependent cytotoxicity that is enhanced by cetuximab. Therefore, the combination of cetuximab and avelumab could result in synergistic activity and could be a strategy for potentiating antitumor activity. The primary endpoint was OS. A post hoc analysis was performed to assess the efficacy of cetuximab plus avelumab according to ctDNA levels; mOS was 11.6 months (95% CI, 8.4–14.8 months); mPFS was 3.6 months (95% CI, 3.2–4.1 months); the ORR was 7.8%; and one patient had CR. Among the 67 patients who were assessed for ctDNA, patients with *RAS*/*BRAF* WT had longer mOS and mPFS when compared to patients with mutated ctDNA (mOS, 17.3 vs. 10.4 months; HR, 0.49; 95% CI, 0.27–0.90; *p* = 0.02 and mPFS, 4.1 vs. 3.0 months; HR, 0.42; 95% CI, 0.23–0.75; *p* = 0.004) [108]. In these trials, the ORR was 0–20%, and the mPFS was approximately 3 months. While these results were not satisfactory, a better trend was observed in patients with no ctDNA mutations than in patients with some mutations. Therefore, evaluation of ctDNA related to resistance could be useful for identifying patients eligible for anti-EGFR mAb rechallenge.

Trials to prospectively evaluate the efficacy of anti-EGFR mAb rechallenge using ctDNA have been conducted. The CHRONOS study, an open-label, single-arm, phase II clinical trial, was the first to prospectively evaluate the efficacy of rechallenge with EGFR inhibitors based on the mutational status of ctDNA. The main inclusion criteria were *RAS*/*BRAF* WT in tissue, CR or PR to anti-EGFR mAbs, progression after the last treatment without anti-EGFR mAbs, and *RAS* and *BRAF* WT and *EGFR* ECDs in ctDNA. The primary endpoint was the ORR. The plasma *RAS* status was measured using ddPCR. Among the patients with no detectable alterations in *RAS*, *BRAF*, and *EGFR* ECDs in ctDNA, eight (30%) achieved PR. In this study, there was no correlation between the aEFI and the probability of response. However, the CHRONOS study showed that patients with a shorter aEFI (within 12 months) responded to anti-EGFR mAb rechallenge. Therefore, the optimal aEFI varied between patients and was not based on a certain period. The results of CHRONOS indicate that selecting patients based on ctDNA would enable the selection of more appropriate candidates for anti-EGFR mAb rechallenge because it would allow the exclusion of resistant cases [109]. PURSUIT is a multicenter, single-arm phase II trial that evaluated the efficacy of anti-EGFR mAb rechallenge in patients with mCRC and plasma *RAS* WT. In the monitoring phase, REMARRY prospectively monitored plasma *RAS* status in patients with *RAS*/*BRAF* WT mCRC following a refractory response to anti-EGFR mAbs. Plasma *RAS* status was measured at disease progression during subsequent therapies, and patients were enrolled in the PURSUIT trial if they tested negative for plasma *RAS*. Other key eligibility criteria included CR or PR to previous anti-EGFR mAb treatment and an interval ≥4 months since the last administration of anti-EGFR mAbs. Plasma *RAS* status was measured using the OncoBEAM^TM^ RAS CRC Kit. The primary endpoint was not met, and the confirmed ORR was 14%. The subgroup analysis showed a significantly higher confirmed ORR in patients with a longer aEFI than in those with a shorter aEFI (>365 vs. <365 days, 44.4% vs. 7.3%, *p* = 0.0037). The aEFI is assumed to be a factor in predicting the effectiveness of the anti-EGFR mAb rechallenge in the PURSUIT trial, as opposed to that in the CHRONOS study. Notably, five patients with plasma *RAS* WT had a confirmed response (ORR, 16%), whereas no response was observed in seven patients with plasma *RAS* mutations (ORR, 0%) (*p* = 0.25) [110]. VELO is a randomized phase II trial to evaluate anti-EGFR mAb rechallenge as a third-line treatment in patients with mCRC. The patients were randomized 1:1 to receive panitumumab plus trifluridine/tipiracil or trifluridine/tipiracil alone. The main inclusion criteria were achieving CR or PR to anti-EGFR mAb as first-line therapy and an aEFI of at least 4 months. Plasma samples were prospectively collected from all patients. Patients who received panitumumab plus trifluridine/tipiracil had a PFS of 4 months, whereas those who received only trifluridine/tipiracil had a PFS of 5 months. In addition, at 6 months, the PFS rate was 38.5% in patients with *RAS*/*BRAF* WT and 13% in those with *RAS*/*BRAF* MT. Thirteen patients (20.9%) had *RAS*/*BRAF* WT at baseline ctDNA analysis [111]. Anti-EGFR mAb rechallenge was investigated in combination with a cyclin-dependent kinase 4/6 inhibitor in a phase II trial; however, the results were not favorable. The 4-month disease control rate was 20%, and the mPFS was 1.8 months [112]. Anti-EGFR mAb rechallenge is well tolerated and not substantially different from anti-EGFR mAb as a first-line therapy. Confirmation of ctDNA mutational status is essential when considering anti-EGFR mAb rechallenge. The optimal interval between the initial administration and re-initiation of anti-EGFR mAb therapy remains unclear.

### 4.2. Ongoing Trials of anti-EGFR Monoclonal Antibody Rechallenge Based on Liquid Biopsy

Several Clinical Trials of anti-EGFR mAb rechallenge are Ongoing (Table 2).

PULSE (NCT03992456) is a randomized, phase II, open-label study designed to compare the OS of panitumumab rechallenge with that of standard-of-care treatment (trifluridine/tipiracil or regorafenib) for patients with mCRC with no resistance mutations, as determined by liquid biopsy. The inclusion criteria are progression after at least four months of treatment with an anti-EGFR mAb and >90 days between the recent administration of anti-EGFR mAb and liquid biopsy. Secondary objectives include comparisons of PFS, ORR, clinical benefit rate, and quality of life, as measured using a linear analog self-assessment questionnaire. A total of 120 patients will be randomized 1:1 to receive panitumumab rechallenge or standard-of-care treatment. This trial will optimize the third-line regimen after the progression of anti-EGFR mAb in patients with *RAS/BRAF* WT mCRC.

PARERE (NCT04787341) is a prospective, open-label, multicenter, phase II study aimed at evaluating the anti-EGFR mAb rechallenge sequence of *RAS*/*BRAF* WT, chemo-refractory mCRC with previous benefit from first-line anti-EGFR-mAb-based treatment according to ctDNA analysis using liquid biopsy. *RAS*/*BRAF* WT ctDNA at the time of screening, at least a PR or SD ≥6 months since the first anti-EGFR-mAb-containing regimen, and at least 4 months between the end of first-line anti-EGFR mAb and liquid biopsy were required. A total of 214 patients were randomized in a 1:1 ratio to receive panitumumab followed by regorafenib after progression or the reverse sequence. The primary endpoint is OS. The secondary endpoints are 1st-PFS, 2nd-PFS, time to failure strategy, ORR, and safety. The results of this study will validate the appropriate placement of anti-EGFR mAb rechallenge in treatment strategies and provide useful knowledge regarding the aEFI.

CAPRI II GOIM (NCT05312398) is an open-label, phase II study investigating the efficacy and safety of a biomarker-driven, cetuximab-based treatment regimen over three treatment lines in patients with *RAS*/*BRAF* WT mCRC at the start of first-line treatment. Patients will be treated with cetuximab in combination with chemotherapy as follows: FOLFIRI (fluorouracil, leucovorin, and irinotecan) plus cetuximab (first-line), FOLFOX plus cetuximab (second-line), and irinotecan plus cetuximab (third-line). If *RAS* and/or *BRAF* mutation status is detected in ctDNA during disease progression, patients will be treated with FOLFOX plus bevacizumab as a second-line therapy or with regorafenib or trifluridine/tipiracil (investigator’s choice) as a third-line therapy. In cases where *RAS*/*BRAF* WT is observed through liquid biopsy at each time point of progression, patients are treated with cetuximab rechallenge in combination with irinotecan. In total, 200 patients will be enrolled. The primary endpoint is the ORR. The secondary endpoints are PFS, OS, AEs, EORTC QLQ C30, and DERMATOLOGY LIFE QUALITY INDEX. This study will reveal the significance of continuous anti-EGFR mAb administration in patients with *RAS*/*BRAF* WT mCRC based on dynamic and longitudinal liquid biopsy assessments of *RAS*/*BRAF* status.

CAVE II GOIM (NCT05291156) is a phase II, open-label, randomized clinical study to assess the efficacy of the combination of avelumab and cetuximab as a rechallenge strategy in patients with *RAS*/*BRAF* WT mCRC who achieved CR or PR after first-line therapy with cetuximab. A total of 173 patients were randomly assigned in a 2:1 ratio to receive either avelumab plus cetuximab or cetuximab alone. Patients with *RAS*/*BRAF* WT on liquid biopsy at screening were enrolled in the study. The primary endpoint is OS. The combination of cetuximab plus avelumab for patients with *NRAS* and *KRAS* WT mCRC was effective in the CAVE trial, a phase II single-arm trial. Cetuximab in combination with avelumab could potentiate antitumor activity as an anti-EGFR mAb rechallenge.

Although plasma samples for liquid biopsy were not collected, a randomized phase III trial, the FIRE-4 study (NCT02934529), is being conducted to evaluate irinotecan plus cetuximab as a third-line therapy in patients with *RAS* WT mCRC. Achieving CR or PR with a PFS of ≥6 months, FOLFIRI plus cetuximab as a first-line treatment, FOLFOX plus bevacizumab as a second-line treatment, and *RAS* WT tumor status were the inclusion criteria. A total of 550 patients were randomized to receive cetuximab rechallenge in combination with irinotecan-based chemotherapy or anti-EGFR-mAb-free treatment. The primary endpoint is OS.

## 5. Neo*RAS*

Approximately 55% of mCRC patients have *RAS* mutations at diagnosis [113]. However, administration of anti-EGFR mAbs is not recommended for patients with *RAS* WT mCRC. Surprisingly, reversal from *RAS* MT to *RAS* WT has recently been reported [114,115,116]. This phenomenon is called Neo*RAS* WT. The mechanism underlying Neo*RAS* WT mCRC remains unclear. In the presence of CRC with a low allele frequency of *RAS* close to the cut-off level, the MAF of *RAS* generally lies below the detection threshold after treatment, which could result in Neo*RAS* WT mCRC (Figure 2). The evolutionary pressure imposed by the tumor microenvironment and treatments leads to pulsatile levels of *RAS* MT clones and negative selection against them [117]. The incidence of Neo*RAS* WT mCRC was estimated as 18.8–83.3% in a recent report [117,118,119,120,121,122]. The variation in the results could be attributed to factors such as the small sample size and lack of consensus on the definition of Neo*RAS* WT mCRC. Decisions pertaining to *RAS* WT based on ctDNA analysis are limited by false negatives from ctDNA analysis. In cases where only *RAS* mutations are analyzed, it is not clear to what extent the detection of *RAS* mutations depends on reversion to *RAS* WT or false negatives. NGS and methylation analyses are useful for confirming or excluding the presence of ctDNA in plasma samples [118]. The detection of at least one somatic mutation other than *RAS* is an indicator of sufficient ctDNA in the sample, and methylation can also be used as a cancer-related biomarker for the amount of ctDNA present in a plasma sample. Among 18 patients with no *RAS*/*BRAF* mutations in plasma ctDNA samples, true *RAS* conversion occurred in 15 patients, as determined by NGS and methylation analysis [123]. Anti-EGFR mAbs are effective in patients with *RAS* WT and could also be effective in patients with Neo*RAS* WT. A pilot study evaluated the efficacy and safety of anti-EGFR mAb plus chemotherapy in patients with Neo*RAS* WT mCRC. The ORR was 55.6% in patients with Neo*RAS* WT mCRC treated with a regimen of cetuximab plus FOLFIRI compared to 42.9% in patients where *RAS* MT ctDNA was detected. The PFS was 13.3 months in patients with Neo*RAS* WT mCRC compared to 3.5 months in patients with *RAS* MT mCRC ctDNA. Therefore, anti-EGFR mAbs may be effective in patients with Neo*RAS* WT mCRC [124]. Several clinical trials have evaluated the efficacy of anti-EGFR mAbs in treating patients with Neo*RAS* WT mCRC.

### Ongoing Trials for the Treatment of NeoRAS Wild-Type Metastatic Colorectal Cancer

Multiple trials for treatments for Neo*RAS* WT mCRC are ongoing (Table 3).

The CETIDYL study (NCT04189055) is a single-arm, phase II study to evaluate the efficacy of cetuximab (Cohort 1) and cetuximab and irinotecan (Cohort 2) as salvage therapies in patients with Neo*RAS* WT mCRC who previously received standard therapies for liver metastasis. Seventy-two patients were enrolled in the study. Patients were initially included in Cohort 1. Inclusion in Cohort 2 will start when the results of Cohort 1 are available. The primary endpoint is the ORR.

The KAIROS study (EudraCT number 2019-001328-36) is a single-arm, phase II study that aims to evaluate the safety and efficacy of cetuximab plus chemotherapy as a second-line treatment in 112 patients with Neo*RAS* WT mCRC. The primary endpoint is the ORR.

The MoLiMoR study (NCT04554836) is a prospective, randomized, phase II study conducted to evaluate the efficacy and safety of FOLFIRI-based first-line therapy with or without intermittent cetuximab for Neo*RAS* WT mCRC. In the FOLFIRI + cetuximab group, treatment was shifted to FOLFIRI at the emergence of *RAS* mutation as well as to FOLFIRI + cetuximab in cases of repeated conversion to *RAS* WT. Key eligibility criteria were true *RAS* MT and left-sided mCRC. A total of 144 patients were randomized to receive FOLFIRI + intermittent cetuximab or FOLFIRI. The primary endpoint is the PFS.

The C-PROWESS study (jRCTs031210565) is a multicenter, single-arm, phase II study investigating the safety and efficacy of panitumumab and irinotecan in 30 patients with Neo*RAS* WT mCRC. The key eligibility criteria are mCRC tissue with *RAS* MT, refractory or intolerant to fluoropyrimidine, oxaliplatin, or irinotecan, and *RAS* WT in ctDNA. The primary endpoint is ORR [125].

CONVERTIX (2017-003242-25) was a single-arm, phase II study to evaluate the efficacy of second-line treatment with panitumumab + FOLFIRI in *RAS* WT mCRC patients who underwent *RAS* MT at the initiation of first-line treatment with FOLFOX plus bevacizumab. However, this study was terminated early because of a lack of eligible patients; 23 patients were screened, but none met the selection criteria according to the abbreviated clinical study report.

## 6. Conclusions

The information that can be extracted from ctDNA could be used to confirm real-time tumor genetic information and to optimize the strategy of chemotherapy regimens, such as anti-EGFR mAb rechallenge and anti-EGFR mAb for Neo*RAS* WT mCRC. The advantages and limitations of ctDNA information should be considered when interpreting these results. The development of technologies to assay ctDNA will provide a basis for personalized medicine and will likely change treatment strategies not only in CRC but also in various types of cancers.

## Figures and Tables

**Figure 1 cancers-15-01473-f001:**
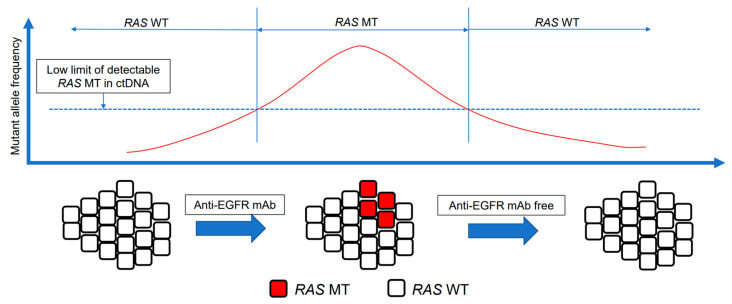
Concept of anti-EGFR mAb rechallenge. The intratumor heterogeneity of mCRC and the dynamism of clonal evolution under the selection pressure of treatment. The emergence of RAS mutations in tumors is a well-recognized mechanism of acquiring resistance to anti-EGFR mAbs. RAS mutations are sub-clonal mutations with a low allele frequency. After withdrawal of anti-EGFR mAbs, chemotherapy restores anti-EGFR mAb sensitivity. RAS MT clones are not detected, and EGFR-sensitive clones are predominant. mAb, monoclonal antibody; EGFR, epidermal growth factor receptor; ctDNA, circulating tumor DNA; MT, mutant; WT, wild-type.

**Figure 2 cancers-15-01473-f002:**
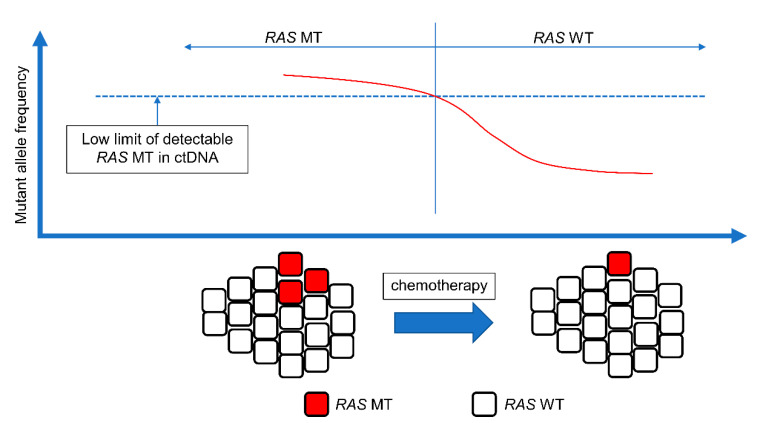
Concept of Neo*RAS* WT. The presence of tumor cells with low *RAS* allele frequency close to the cutoff level. The mutant allele frequency of *RAS* is below the detection threshold after the treatment, resulting in Neo*RAS* WT mCRC. mCRC, metastatic colorectal cancer; ctDNA, circulating tumor DNA; MT, mutant; WT, wild-type.

**Table 1 cancers-15-01473-t001:** Previous reports of anti-EGFR monoclonal antibody rechallenge.

Study	Phase	Assay	Number	Primary Endpoint	Treatment	ORR (%)	mPFS (Months)	mOS (Months)
CRIKET [105]	Single-arm phase II	Droplet digital PCR	28 (ctDNA: 25)	ORR	CET + IRI	21	3.4	9.8
E-Rechallenge [106]	Single-arm phase II	Droplet digital PCR	33 (ctDNA: 24)	ORR	PANI + IRI	15.2	2.9	8.7
JACCRO CC-08/09AR [107]	Single-arm phase II	BEAMing (OncoBEAM^TM^ RAS CRC KIT)	16	3-month PFS	CET + IRI (08) PANI + IRI (09)	0	2.4	8.2
CAVE [108]	Single-arm phase II	Quantitative PCR	77 (ctDNA: 67)	OS	Avelumab + CET	7.8	3.6	11.6
CHRONOS [109]	Single-arm phase II	Droplet digital PCR	27	ORR	PANI	30	N/A	12.8
PURSUIT [110]	Single-arm phase II	BEAMing (OncoBEAM^TM^ RAS CRC KIT)	50	ORR	PANI + IRI	14	3.6	N/A
VELO [111]	Randomized phase II	Real-time PCR (Idylla^TM^)	62	PFS	PANI + TAS102 vs. TAS102	9.7	4.0 vs. 2.5	N/A

EGFR: epidermal growth factor receptor; PCR: polymerase chain reaction; ctDNA: circulating tumor DNA; ORR: overall response rate; mOS: median overall survival; mPFS: median progression-free survival; CET: cetuximab; PANI: panitumumab; IRI: irinotecan; TAS102: trifluridine/tipiracil.

**Table 2 cancers-15-01473-t002:** Ongoing trials of anti-EGFR monoclonal antibody rechallenge based on liquid biopsy.

Study	Phase	Number	Assay	Estimated Enrollment	Treatment	Primary Endpoint	Study Completion Date
PULSE	Randomized phase II	NCT03992456	NGS (Guardant360^®^ assay)	120	PANI vs. TAS102 or regorafenib	OS	7 October 2023
PARERE	Randomized phase II	NCT04787341	NGS (Oncomine^TM^ Colon cfDNA Assay)	214	PANI → regorafenib vs. regorafenib → PANI	OS	15 June 2024
CAPRI II GOIM	Single-arm phase II	NCT05312398	NGS	200	The therapeutic algorithm by liquid biopsy assessment of *RAS/BRAF* status	ORR	15 June 2026
CAVE II GOIM	Randomized phase II	NCT05291156	NGS (FoundationOne^®^ Liquid)	173	CET + avelumab vs. CET	OS	1 July 2025

EGFR: epidermal growth factor receptor; NGS: next-generation sequencing; ORR: overall response rate; OS: overall survival; CET: cetuximab; PANI: panitumumab; TAS102: trifluridine/tipiracil.

**Table 3 cancers-15-01473-t003:** Ongoing trials for the treatment of Neo*RAS* wild-type metastatic colorectal cancer.

Study	Phase	Number	Assay	Estimated Enrollment	Treatment	Primary Endpoint	Study Completion Date
CETIDYL	Single-arm phase II	NCT 04189055	Real-time PCR (Idylla^TM^)	72	CET (Cohort1) or CET + IRI (Cohort2)	ORR	7/1/2023
KAIROS	Single-arm phase II	EudraCT 2019-001328-36	Real-time PCR (Idylla ^TM^)	112	CET + chemotherapy	ORR	N/A
MoLiMoR	Randomized phase II	NCT 04554836	Droplet digital PCR	144	FOLFIRI + intermittent CET vs. FOLFIRI	PFS	10/1/2024
C-PROWESS [118]	Single-arm phase II	jRCTs 031210565	BEAMing (the OncoBEAM ^TM^ RAS CRC KIT)	30	PANI + IRI	ORR	1/1/2025

PCR: polymerase chain reaction; ORR: overall response rate; PFS: progression-free survival; CET: cetuximab; PANI: panitumumab; IRI: irinotecan; FOLFIRI: fluorouracil + leucovorin + irinotecan.

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
