# Peer review of "Circulating Tumor DNA: The Dawn of a New Era in the Optimization of Chemotherapeutic Strategies for Metastatic Colo-Rectal Cancer Focusing on RAS Mutation"

_cancers, 2023, doi:10.3390/cancers15051473_

Round 1

Reviewer 1 Report

The authors presented a comprehensive review entitled “ Circulating tumor DNA: The dawn of a new era in the optimization of chemotherapeutic strategies for metastatic colorectal cancer focusing on RAS mutation”

The topic is relevant. The manuscript provides updated information on the role of ctDNA in CRC

However, some concerns should be addressed to improve overall quality.

The manuscript should be more accurate. 

The introduction should 

a) include a concise section on RAS (including K and NRAS) and BRAF 

b) only cover CRC.

"Approaches for ctDNA detection" should draw a clear distinction between informed and non-informed approaches to ctDNA detection. Providing a table or figure would be helpful

 If the manuscript is focused on RAS, the review should describe available data on RAS and eventually BRAF. The inclusion of topics such as MRD and early detection, which do not directly pertain to the main focus, should be reconsidered.

Moreover, "Assessment of prognosis" refers to the topic in a generic way; it includes a limited number of trials. A more comprehensive description of the prognostic impact of ctDNA based on novel data would be valuable 

Reviewer 2 Report

The authors present a thorough and broad presentation over the field of circulating tumour DNA. I believe this is of interest to  many researchers in the field of colo rectal cancer. The presentation seems balanced presenting pros and cons for different methods. The presentation could be improved by adding more pictures and/or tables to balance the mass of text.

Reviewer 3 Report

In this manuscript, Authors present a large review about the role of liquid biopsy in colorectal cancer, highlighting its advantages and limits. Nowadays, liquid biopsy is predominantly used in lung cancer but several trials have already demonstrated its feasibility in colon cancer too. Unfortunately, international guidelines do not still acknowledge it as a standard technique to establish the mutational status of patients undergoing treatments for the metastatic disease.

However, liquid biopsy is very helpful during treatment as reported by Authors to relieve changing of the mutational status for the increase of mutant clones in response to the selective pressure exerted by anti-EGFRs.

Considering the manuscript for publication, I would suggest to cite the role of intermittent strategies and reintroduction of anti-EGFRs, recently presented in 2022, and to underline the important negative prognostic role represented by BRAF V600 mutations (the introduction reports the negative role of RAS mutations, only).   
